# Diversity of Clinically Relevant Outcomes Resulting from Hypofractionated Radiation in Human Glioma Stem Cells Mirrors Distinct Patterns of Transcriptomic Changes

**DOI:** 10.3390/cancers12030570

**Published:** 2020-03-01

**Authors:** Darius Kalasauskas, Maxim Sorokin, Bettina Sprang, Alhassan Elmasri, Sina Viehweg, Gabriela Salinas, Lennart Opitz, Margret Rave-Fraenk, Walter Schulz-Schaeffer, Sven Reiner Kantelhardt, Alf Giese, Anton Buzdin, Ella L. Kim

**Affiliations:** 1Laboratory for Experimental Neurooncology, Clinic for Neurosurgery, Johannes Gutenberg University Medical Centre, 55131 Mainz, Germany; Darius.Kalasauskas@unimedizin-mainz.de (D.K.); bettina.sprang@unimedizin-mainz.de (B.S.); alhassan-96@hotmail.de (A.E.); sina.viehweg@gmail.com (S.V.); 2Clinic for Neurosurgery, Johannes Gutenberg University Medical Centre, 55131 Mainz, Germany; sven.kantelhardt@unimedizin-mainz.de; 3Shemyakin-Ovchinnikov Institute of Bioorganic Chemistry, 117997 Moscow, Russia; sorokin@oncobox.com (M.S.); buzdin@oncobox.com (A.B.); 4I.M. Sechenov First Moscow State Medical University, 119991 Moscow, Russia; 5Omicsway Corp., Walnut, CA 91789, USA; 6NGS Integrative Genomics Core Unit (NIG), Institute for Human Genetics, University Medical Centre, 37077 Göttingen, Germany; gsalina@gwdg.de (G.S.); lennart.opitz@fgcz.ethz.ch (L.O.); 7Department of Radiotherapy and Radiooncology, University Medical Centre, 37077 Göttingen, Germany; margret.rf@gmail.com; 8Department of Neuropathology, Saarland University Medical Centre, 66421 Homburg, Germany; Walter.Schulz-Schaeffer@uks.eu; 9OrthoCentrum Hamburg, Department of Tumor Spinal Surgery, 20149 Hamburg, Germany; alf.giese1@gmail.com; 10Oncobox ltd., 121205 Moscow, Russia; 11Moscow Institute of Physics and Technology (National Research University), 141700 Moscow, Russia

**Keywords:** glioblastoma, glioma stem cells, hypofractionated radiation, radioresistance

## Abstract

Hypofractionated radiotherapy is the mainstay of the current treatment for glioblastoma. However, the efficacy of radiotherapy is hindered by the high degree of radioresistance associated with glioma stem cells comprising a heterogeneous compartment of cell lineages differing in their phenotypic characteristics, molecular signatures, and biological responses to external signals. Reconstruction of radiation responses in glioma stem cells is necessary for understanding the biological and molecular determinants of glioblastoma radioresistance. To date, there is a paucity of information on the longitudinal outcomes of hypofractionated radiation in glioma stem cells. This study addresses long-term outcomes of hypofractionated radiation in human glioma stem cells by using a combinatorial approach integrating parallel assessments of the tumor-propagating capacity, stemness-associated properties, and array-based profiling of gene expression. The study reveals a broad spectrum of changes in the tumor-propagating capacity of glioma stem cells after radiation and finds association with proliferative changes at the onset of differentiation. Evidence is provided that parallel transcriptomic patterns and a cumulative impact of pathways involved in the regulation of apoptosis, neural differentiation, and cell proliferation underly similarities in tumorigenicity changes after radiation.

## 1. Introduction

Glioblastoma (GB) is the most malignant form of astrocytic tumors in adults with dismal prognosis [1]. GB recurrence after (or even under) cytotoxic therapy and alkylating chemotherapy is virtually inevitable and poses a major challenge for improving clinical outcomes of patients with GB [2,3]. For newly diagnosed GB (ndGB), hypofractionated ionizing radiation (frIR) is the component of standard therapy consisting of surgical resection followed by the concomitant treatment with frIR and alkylating chemotherapy [4]. For recurrent GBs (recGB), no effective therapies are currently available [2,3]. Despite intensive efforts to improve the efficacy of cytotoxic monotherapies for recGBs no level one evidence could be achieved so far.

The high degree of genetic and cellular complexity of these tumors is the biological basis for GBs resistance to cytotoxic and targeted therapies. On the molecular level, there is a multiplicity of genomic aberrations in numerous pathways forming an interconnected signaling network [5]. The current paradigm of gliomagenesis and malignant progression is centered on so-called glioma stem cells (GSCs) implicated as the most tumorigenic type of glioma cells driving GBs growth before and after therapy [6,7,8,9]. Owing to their unique biological properties such as an unlimited capacity to self-renew and inherent plasticity, GSCs are capable of maintaining the tumor homeostasis and perpetuating intratumoral heterogeneity, which is as an important factor contributing to GBs resistance to cytotoxic and targeted therapies [10,11,12].

Undifferentiated GSCs comprise a relatively rare population of cells residing in specialized niches distributed non-randomly throughout the tumor [13]. The inherent plasticity and ability of GSCs to interconvert between different cell states during tumor progression poses a challenge to delineating the exact mechanisms underlying GSC-mediated resistance to cytotoxic treatments. Adding a further level of complexity, there exist different types of GSCs that are not uniform but constitute a distinct yet phenotypically and molecularly heterogeneous compartment resembling the hierarchical structure of neural lineages [14,15,16].

While there is a general consensus is that GSCs play important roles in driving recurrent tumor growth after cytotoxic therapies, there exist considerable controversies about the effects of cytotoxic treatments on GSCs ability to propagate tumor growth and the mechanisms involved. On the one hand, GSCs have been shown to have increased levels of DNA repair, which renders cells less vulnerable to survival-prohibiting effects of DNA damaging cytotoxic treatments such as ionizing radiation [17]. On the other hand, there is also evidence that irradiation may reduce the tumorigenic potential of GSCs presumably through the induction of cell differentiation [18].

While the reasons for existing discrepancies remain unclear, a considerable heterogeneity in experimental treatments and experimental endpoints used in different studies might have a significant impact. This is especially evident across studies that have addressed the effects of ionizing radiation in GSCs. Experimental regimens commonly used to characterize GSCs responses to radiation consist of a single exposure to escalating radiation doses that deviate from the hypofractionated regimen used for radiotherapy of GB (daily fractions at an invariable dose of 2–2.5 Gy for a total dose of around 60 Gy) [4]. It is also uncertain to what extent surrogate endpoints used in laboratory investigations using GSCs adequately reflect changes in clinically meaningful endpoints. Previous investigations in GSCs have primarily been focused on molecular or cellular responses occurring within a relatively short period of time (several hours to several days) after experimental treatments. In contrast, there is a paucity of information regarding the long-term changes invoked by clinically relevant regimens of radiation in GSCs. Considering that recurrent GBs re-grow over a time span of several weeks to several months after radiochemotherapy the characterization of long-term effects of hypofractionated radiation can provide important insights into the molecular and cellular mechanisms underlying post-therapy growth of GB.

In this study, long-term outcomes of clinically relevant doses of fractionated radiation were investigated in a panel of human GSCs using an integrated approach based on parallel assessments of the tumor-propagating capacity, stemness properties and gene expression patterns.

## 2. Results

### 2.1. GSC Models Used in the Study

A panel of human GSCs used in this study was isolated from newly diagnosed GBs and characterized for neural lineage markers and stem cell markers in our previous studies [19,20,21,22,23]. In brief, the abundant expression of the neural stem/progenitor cell marker nestin is a common characteristic of all GSCs analyzed in this study (Appendix A). In contrast, an astrocyte marker glial fibrillary acidic protein (GFAP) shows variable expression patterns in different lines and under different culture conditions (Appendix A). Considering that GFAP is a phenotypic marker of astrocyte differentiation, the heterogeneity of GFAP phenotypes suggest varying degrees of the differentiation capacity. To test this assumption, we estimated the proportion of GFAP positive cells in relation to differentiation. To visualize morphological changes, a hallmark of cell differentiation, GSCs were analyzed by immunocytochemical staining as illustrated in Figure 1a. The results obtained for different lines are summarized in Figure 1b and show that an exposure to differentiation-inducing condition (bFGF and EGF withdrawal) leads to a significant increase in the proportion of GFAP-expressing cells in some (but not all) GSCs (Figure 1b), concomitantly with the acquisition of morphology characteristic of differentiated cells (Figure 1a and Appendix A).

Such a pattern (called hereafter GFAP^Ind^) is indicative of the ability to differentiate. To further validate this conclusion, we evaluated the proliferative activity of GSCs under self-renewal promoting or differentiation-inducing condition. To that end, the proportion of proliferating cells was estimated by cells immunostaining for Ki67 as illustrated in Appendix A. The results showed that exposure of GFAP^Ind^ GSCs to differentiation-inducing condition induces proliferative decline, a hallmark of cell differentiation (Figure 1c).

We undertook a detailed characterization of GSCs with respect to the fundamental characteristics of cancer stem cells such as self-renewal potential and tumor-propagating capacity. Self-renewal was evaluated by using the ELDA (extreme limiting dilution assay) method, which enables us to estimate the fraction of stem cells in the state of self-renewal [24]. The results showed that all GSCs used in the study possess the propensity to self-renew albeit to varying degrees (Table 1, SCF). To determine if the differing degrees of self-renewal correlate with the tumor-promoting capacity we made use of the orthotopic xenograft model. All GSCs used in the study proved to be highly tumorigenic, as evidenced by a tumor take of 100% (data not shown), and capable of giving rise to aggressive brain tumors with irregular and invasive morphology characteristic of GB (Appendix A). Comparison of growth rates of tumors driven by different GSC lines revealed marked differences (Table 1, TGR). Quite the contrary to the general assumption that self-renewal reflects the degree of tumor-propagating capacity in GSCs, there was a lack of correlation between tumor growth rates and self-renewal capacity (Table 1, Figure 2). Notably, GFAP^Ind^ xenografts showed a tendency towards slower growth and lower proliferation indices (Ki67^in vivo^) compared to GFAP^Const^ xenografts (Table 1).

There were also marked changes in GFAP expression patterns between GSCs propagating in vitro or in vivo. Phenotypic changes were especially striking in tumors initiated by self-renewing GFAP^Ind^ GSCs. in vitro, GFAP^Ind^ GSCs express low levels of GFAP in the state of self-renewal (Figure 1a and Appendix A) but in vivo they show a dramatic shift towards GFAP+ phenotype characteristic of the differentiated cell state (Figure 3a). In contrast, tumors initiated by GFAP^Const^ GSCs show massive loss of GFAP expression suggesting a reversion to a less differentiated phenotype (Figure 3b). The ability of GFAP^Ind^ GSCs but not GFAP^Const^ GSCs to undergo spontaneous conversion in vivo towards a less proliferative differentiated state explains the slower growth of GFAP^Ind^ xenografts compared to GFAP^Const^ xenografts (Table 1, Figure 2). In contrast to the variable expression of GFAP, nestin expression shows similar patterns in vitro and in vivo (Appendix A, Appendix A).

### 2.2. Proliferative Activity of Differentiating GSCs Contributes to Tumor Growth after Radiation

We next sought to evaluate the consequences of fractionated radiation (fr-IR) to the tumor-propagating capacity and proliferative potential of GSCs. Tumor-matched non-radiated GSC and their fr-IR treated counterparts (designated as “GSC_IR”) were compared as depicted in Appendix A. Comparative survival analyses revealed that fr-IR treatment leads to non-uniform changes in the tumor-propagating capacity (TPC) in different GSCs showing either a reduction, augmentation or no change in the degree of tumorigenicity after radiation (Figure 4). According to the type of change in TPC after fr-IR, GSCs were designated as rad-TS (radiation-associated tumor suppression), rad-TP (radiation-associated tumor promotion), or Inv (invariable).

We next addressed the effects of fr-IR treatment on the proliferative activity. To that end, the rates of BrdU incorporation (BIR) were compared between isogenic GSCs either non-radiated or frIR-treated. Keeping in mind that GFAP^Ind^ and GFAP^Const^ GSCs differ in the ability to undergo spontaneous differentiation during tumor growth (Figure 3) proliferative capacity was evaluated under either self-renewal promoting or differentiation-inducing condition. In all GSC pairs tested, significant proliferative changes invoked by fr-IR were found under either self-renewal promoting or differentiation-inducing condition (Table 2). Interestingly, there was an association between the pattern of post-radiation change in BIR and cellular state. In GFAP^Ind^ GSCs, proliferative changes induced by fr-IR are most profound under differentiation-inducing condition, whereas in GFAP^Const^, proliferative changes after fr-IR appear to occur primarily in the state of self-renewal (Table 2).

### 2.3. Diverse Patterns of Changes in Gene Expression Mirror the Diversity of Tumor Growth Patterns after fr-IR Treatment

We next sought to determine if there are common molecular traits associated with particular patterns of change in the tumor-propagating potential after fr-IR. To that end, we conducted a pairwise comparison of gene expression in six pairs of tumor-matched GSCs, either non-radiated or fr-IR treated, representing different patterns of tumorigenicity changes after fr-IR (Figure 4). Differential expression analyses revealed significant changes in a large number of genes analyzed (Gene Expression Omnibus GSE140746). Cross-comparison of differentially expressed genes (DEGs) identified in individual GSC/GSC_IR pairs showed varying degrees of overlap but no common DEGs that would be shared across all six pairs (Figure 5a).

The most common DEG identified in five out of six pairs analyzed was gene *has2* coding for hyaluronan synthase 2, an enzyme implicated in the regulation of cell invasion/migration/adhesion in GB (Figure 5a, bottom). Principal component analysis (PCA, Figure 5b) revealed a high degree of transcriptional diversity across different GSC lines. Heatmap build for differentially expressed genes (n = 1989, FDR < 0.05 and |log_2_(FC)| ≥ 1) revealed co-clustering of rad-TS cell lines (#1043 and #1051) but not among GSCs that follow rad-TP pattern (Figure 5c). In addition to gene expression analysis we performed cross-comparison of Gene Ontology terms (GO) for corresponding DEGs (Figure 6). GO analysis revealed the biological processes altered in different GSCs. The results of GO analysis are presented in Appendix A.

To analyze transcriptomic patterns at the level of signaling pathways, we applied Oncobox, a bioinformatic method that enables to quantitatively estimate pathways activities calculated from expression levels of genes relevant for a particular pathway and expressed in so-called pathway activation level scores (PAL/PAS) [25]. PAL scoring conducted for 3125 pathways from the Oncobox database [25] revealed oncopathways affected by fr-IR in different GSCs (Appendix A). In all GSCs/GSC_IR pairs analyzed, Erk5, TGF-beta and mitochondrial apoptosis pathway were most significantly different between non-radiated and radiated GSCs (Figure 7).

There was a striking concordance between the direction of change (up or down) in the three pathways, their biological functions, and experimentally determined outcomes after fr-IR treatment. The non-uniformity of radiation-induced changes in Erk5 and TGF-beta pathways is especially intriguing considering that these pathways have important roles in gliomas and glioma stem cells with Erk5 being a promoter of neural differentiation [26,27,28,29] and TGF-beta a major glioma pathway [30,31,32]. Of special interest is the finding of an inverse correlation between radiation-induced changes in the tumor-propagating potential of GSCs and changes in Erk5 pathway activity: While augmented in radiosensitive GSCs (pattern rad-TS) Erk5 pathway was inhibited in radioresistant GSCs (patterns rad-TP and Inv) (Figure 7). Considering essential roles of Erk5 pathway in promoting neural differentiation this suggests an association between radiation impacts on GSCs ability to differentiate and tumorigenicity changes in radiated GSCs. Parallel changes in Erk5, TGF-beta and apoptotic pathways imply that the ultimate fate of radiated GSCs is determined by cumulative impacts of radiation on signaling pathways involved in the regulation of cell death, differentiation and proliferation.

## 3. Discussion

In this study, the long-term effects of hypofractionated radiation have been investigated in a panel of phenotypically distinct GSCs derived from newly diagnosed glioblastoma. We provide evidence that fr-IR treatment leads to non-uniform changes in the tumor-propagating capacity, proliferative activity, and transcriptomic patterns. In agreement with the general hypothesis that GSCs are primarily responsible for maintaining tumor growth after (or under) radiochemotherapy, our data reveal that the tumor-promoting capacity in some (but not all) types of GSCs can be significantly augmented by fr-IR treatment. We also provide evidence for the existence of GCSs that are actually radiosensitive and susceptible clinically relevant regimens of radiation. Our results reconcile seemingly discordant conclusions on the effects of radiation on GSCs tumorigenicity and underscore the need for considering the diversity of radiation responses across different types of GSCs.

Our study provides novel insights into the relationship between self-renewal, differentiation and tumor-propagating capacity in human GSCs. While it is inarguable that propensity to self-renew is the fundamental characteristic of GSCs, there are divergent views on the role of self-renewal as the primary factor in determining the degree of tumor-propagating capacity or radioresistance in GSCs. On the one hand, self-renewal propensity is widely used as a surrogate measure of the tumor-propagating potential of GSCs. But on the other hand, there is evidence that self-renewal while essential for the maintenance of the undifferentiated state in GSCs does not predict their tumor promoting potential [33]. Data obtained in the current study are consistent with the latter view because: (1) we find no correlation between self-renewal capacity and growth/proliferation rates in GSC-driven tumors (Table 1); (2) radiation-induced changes in self-renewal activity do not parallel changes in the tumor-propagating capacity (Figure 8).

Consistent with the view that GSCs retain the ability proliferate and maintain tumor growth after the exit from self-renewal [33,34,35,36] our data suggest a relationship between proliferative changes at the onset of differentiation and changes in tumorigenicity of radiated GSCs (Table 2). Concordant with the general notion that aberrant differentiation plays important roles in the maintenance of cancer stem cells and their ability to promote tumor growth we find that radiation invokes significant changes in activities of pathways that control cell stemness, proliferation and survival. Of special interest are Erk5 and TGF-beta pathways that have distinct patterns of specialization in the regulation of neuro-/astrocytogenesis. Erk5 pathway while essential for neuronal differentiation is dispensable for astroglial differentiation [26]. TGF-beta exerts a broad range of effects including promotion of astrocyte differentiation, neurogenesis and modulation of neuronal positioning [37]. Notably, Erk5 and TGF-beta pathways show opposite patterns of changes after radiation treatment in GSC sets analyzed in our study (Figure 7). Although the generality of such an inverse relationship needs to be validated in larger sample sets it is tempting to hypothesize that the balance between Erk5 and TGF-beta pathways activities is an important factor determining cellular outcomes from radiation in GSCs. Our data also reveal a considerable diversity of patterns of radiation-induced changes in the mitochondrial apoptotic pathway. In conjunction with the remarkable pleiotropy of TGB-beta and Erk5 pathways the non-uniformity of radiation-associated changes in the apoptotic signaling predicts that GSCs responses to hypofractionated radiation might also be non-uniform. Concordant with the prediction our study provides evidence that fractionated radiation of GSCs can invoke a broad spectrum of outcomes ranging from inhibition, promotion or negligible effect on the tumor-propagating capacity.

## 4. Materials and Methods

### 4.1. Cell Culture and Cell Based Assays

Human GSCs isolated from newly diagnosed GB as described in [19] were maintained in NeuroBasal medium supplemented with the B27 component (Invitrogen Life technologies Baden-Württemberg, Germany), fibroblast growth factor (FGF) and epidermal growth factor (EGF) (10 ng/mL and 20 ng/mL, respectively), (Biochrom GmbH, Berlin, Germany). All the lines used in this study were authenticated at passage 4–6 by using array-based comparative genomic hybridization (aCGH). Cells passaged for not more than 10 passages after authentication were used for in vitro and in vivo analyses described in this study. Self-renewal capacity was evaluated by using the extreme limited dilution assay (ELDA, http://bioinf.wehi.edu.au/software/elda/). In brief, cells were seeded in 24-well plates at a clonal cell density in the range of 0.625 to 10 cells/mL and incubated for 4–6 weeks to allow for clonal spheres formation. Spheres were counted under microscope. Stem cell frequency was calculated using the ELDA software [Hu, 2009 #186]. To assess the differentiation capacity, 30,000–50,000 cells were seeded on glass coverslips pre-coated with poly-l-ornithine hydrobromide (15 µg/mL, Sigma Aldrich, Munich, Germany) and cultured for indicated time points either in the presence or absence of self-renewal promoting factors bFGF and EGF. After 7–10 days incubation, cells were fixed with 4% paraformaldehyde/ PBS (Merck KGaA, Darmstadt, Germany) and assessed for lineage specific markers nestin (Abcam ab22035) or GFAP (Dako Z0334) by immunofluorescence staining. Nuclear counterstaining was performed by using 4,6-diamidino-2-phenylindole (DAPI, Sigma). Antibodies specificity was confirmed by staining of cells with secondary antibodies alone. Cell nuclei were counted by using ImageJ. The proportion of stained cells was determined as a percentage of total nuclei counted in at least four microscopic fields per coverslip. A minimum of 200 cells per condition were counted. All immunostaining experiments were repeated three times.

### 4.2. Cells Irradiation

Hypofractionated radiation regimen consisted of seven fractions of 2 Gy of X-rays generated by a Gulmay RS225 GS014 X-ray machine (Gulmay Medical Ltd., Camberley, UK) at a current of 200 kV, a voltage of 15 mA, and a dose rate of 1 Gy/min. GSC spheres were triturated one day before radiation treatment. Immediately after irradiation cells were placed back to the incubator for recovery and propagation until the next radiation treatment as depicted in Appendix A. Except for irradiation, control cultures were handled identically in parallel. After seven consecutive fractions of radiation or mock-treatments isogenic pairs of mock-treated and radiation-treated GSCs were expanded and used for comparative in vitro, in vivo and in silico analyses.

### 4.3. Animal Experiments and Immunohistochemistry

An orthotopic murine model was used to evaluate the tumorigenic potential of glioma stem cells. Animal experiments were conducted at the Translational Animal Research Center (TARC) of the Johannes Gutenberg University Medical Center with the approval by the State Office of chemical investigations of Rhineland-Palatinate, Mainz, Germany (permission #23 177-07/G12-1-020). Immunodeficient mice (stain NMRI, females, 5–6 weeks old) were obtained from a commercial supplier (Charles River Europe). Mice were maintained in accordance with the guidelines and policies for animal experimentation, housing and care as documented in the European Convention for the Protection of Vertebrates Used for Scientific Purposes. For intracranial implantation, single cell suspensions were prepared from glioma stem cells propagated under self-renewal condition (NeuroBasal medium supplemented with bFGF and EGF) by using a combined enzymatic trypsin/mechanical trituration. Cells were washed twice in PBS, re-suspended in PBS at 3 × 10^4^ cells/µL and evaluated for viability using trypan blue staining. For implantation, 3 µL of cell suspension were injected into the caudato-putamen of the right hemisphere using a stereotactic frame (TSE Systems, Bad Homburg, Germany) using the following stereotactic coordinates: 1 mm (anteroposterior axis), 3 mm (lateromedial axis), and 2.5 mm (vertical axis), in reference to the bregma. Mice were at the first manifestation of neurological symptoms related to brain tumor. For immunohistochemistry, paraffin-embedded brains were sectioned at 1–3 µm thickness.

### 4.4. Microarrays and Bioinformatics Analyses

Total RNA was isolated using Trizol (Invitrogen). Quality of RNA samples was evaluated using the Agilent 2100 Bioanalyzer (Agilent Technologies, Santa Clara, CA, USA) microfluidic electrophoresis. For microarray analysis, 0.3 μg of total RNA was used for cDNA synthesis using the WT Target Labeling and Control Reagents (Affymetrix, Santa Clara, CA, USA). After the cleanup using the GeneChip^®^ Sample Cleanup module (Affymetrix), in vitro transcription was performed using the WT Target Labeling Kit (Affymetrix) followed by purification of the reaction using the GeneChip^®^ cRNA Sample Cleanup Module (Affymetrix) and quantification with the NanoDrop ND-1000 (Nanodrop (Thermo), Wilmington, DE, USA). 5.5 μg of total ssDNA was enzymatically fragmented and labeled by biotinylation using the WT Labeling Kit (Affymetrix). Biotinylated ssDNA fragments were hybridized onto the GeneChip^®^ Human Gene 1.0 ST Array (Affymetrix) according to the manufacturer’s instructions. Affymetrix AGCC Software (version 2.0) (Affymetrix) was used for the extraction of intensity data followed by data analyses using the affy [38] and Limma package [39] of Bioconductor [40] as described [41]. For inter-array normalization quantile-normalization was applied to the log2-transformed intensity values [42]. Differential gene expression analysis was performed by empirical Bayes moderation of the standard errors of the estimated values [43]. *p*-values obtained from the moderated t-statistic were corrected for multiple testing using the Benjamini–Hochberg method [44]. Gene expression data are deposited in Gene Expression Omnibus database with the accession number GSE140746. Cross-comparison of genes expressed differentially in different GSC lines after radiation and preparation of textual and graphical outputs were performed by using the Whitehead BaRC (http://jura.wi.edu) and the Venn Diagramm (http://bioinformatics.psb.ugent.be) software. Pathway analyses were performed by using the Oncobox software described in [45]. For generation of the Oncobox database, which in total consists of 3125 molecular pathways open access pathway catalogues BioCarta [doi:10.1089/152791601750294344], KEGG OC [doi:10.1093/nar/gks1239], [46], Reactome [doi:10.1093/nar/gkn653], [47] and PID [doi:10.1093/nar/gkv1351], [48] were used.

## 5. Conclusions

Treatment with clinically relevant doses of hypofractionated radiation has a direct and significant impact on the tumor-propagating potential of human GSCs. The diverse spectrum of outcomes resulting from radiation in different types of GSCs encompasses an augmentation, reduction or no change in the degree of tumorigenicity. Radiation-induced changes in GSCs tumorigenicity correlate with proliferative changes at the onset of differentiation but not with the change in self-renewal activity. Parallel transcriptomic patterns shaped by the cumulative impact of pathways involved in the regulation of apoptosis, cell differentiation, and proliferation underly the similarities in tumorigenicity changes after radiation. Combinatorial targeting of proliferation, apoptosis and cell differentiation alongside with radiation may be needed to overcome radioresistance mediated by GSCs.

## Figures and Tables

**Figure 1 cancers-12-00570-f001:**
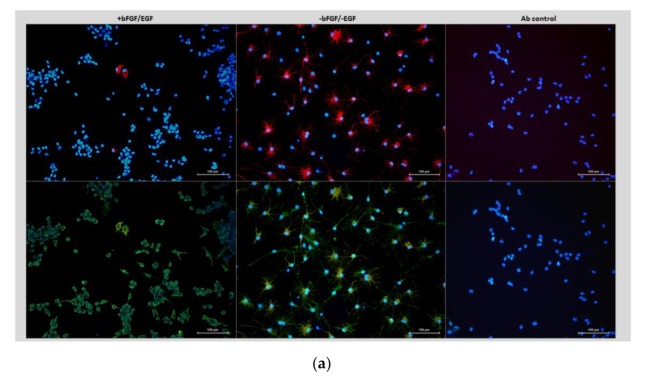
Assessments of GFAP expression and proliferation in different glioma stem cell (GSC) lines. (**a**) Immunocytochemical staining for GFAP (red) and nestin (green) in GSC line #1095 under self-renewal promoting (+bFGF/+EGF) or differentiation-inducing (-bFGF/-EGF) condition. Nuclear counterstaining by DAPI (4’,6-Diamidino-2-Phenylindole, Dihydrochloride). Antibody control panels show samples stained with secondary antibodies (Alexa555 or Alexa488) in the absence of primary antibodies. Magnification 20×. Scale bar, 100 µm. (**b**) Summary of GFAP assessments in a panel of GSC lines. (**c**) Summary of Ki67 assessments in a panel of GSC lines. Data shown as mean ± SEM (standard error of the mean) for three experiments in each group. *** *p* < 0.001, ** *p* < 0.01, * *p* < 0.05, unpaired Student’s *t*-test. “ns”, not significant.

**Figure 2 cancers-12-00570-f002:**
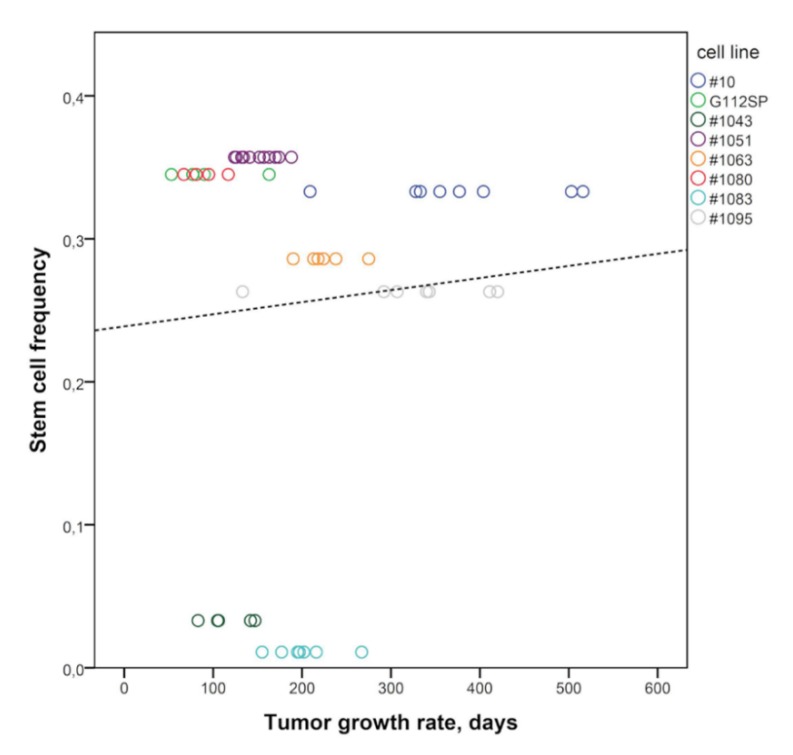
Lack of correlation between the frequency of self-renewing cells and rate of tumor growth driven by GSCs. Circles correspond to the time to individual mice death from a GSC-induced brain tumor. Different GSCs lines are color-coded. Pearson correlation analysis (r = 0.074, *p* = 0.60). Dotted line corresponds to the trend line.

**Figure 3 cancers-12-00570-f003:**
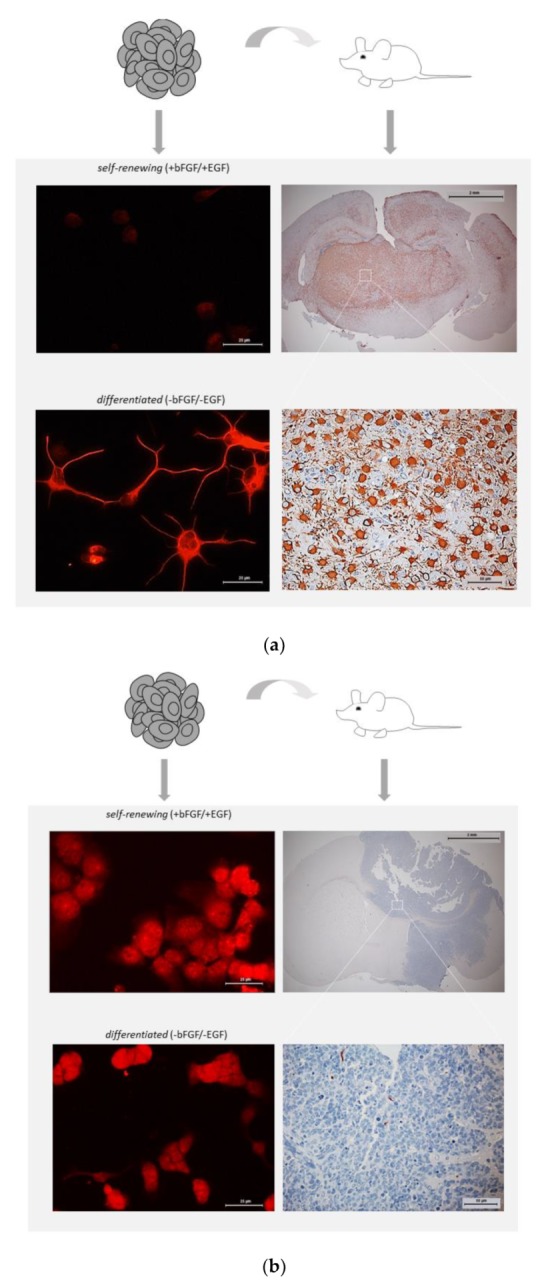
Comparative assessments of GFAP expression in vitro and in vivo. (**a**) GFAP^Ind^ or (**b**) GFAP^Const^ GSCs maintained exclusively under self-renewal promoting condition were implanted into the brain of nude mice to generate tumors. The right panels show immunohistochemical staining of xenograft tumors. Magnification 1.6× (top panels, scale bars 2 mm) and 40× (bottom panels, scale bars 50 µm). For comparison, GFAP patterns of GSCs cultured under self-renewal promoting (**top**) or differentiation-inducing (**bottom**) condition is shown in the left panels. Immunocytochemical staining for GFAP. Magnification 40×, scale bars 25 µm.

**Figure 4 cancers-12-00570-f004:**
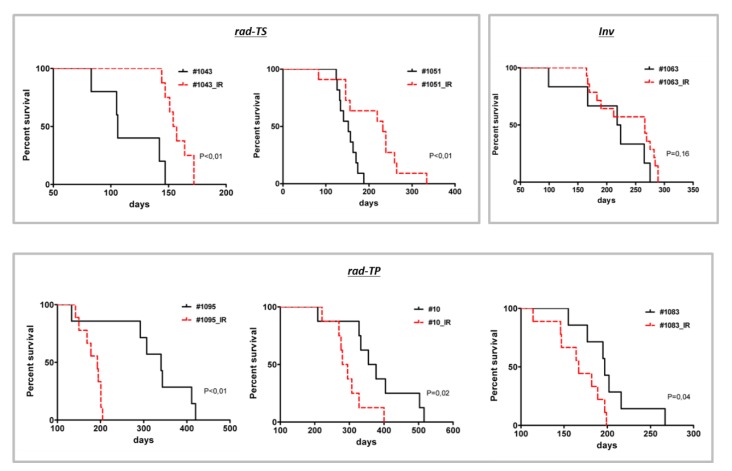
Diverse patterns of change in the tumor-propagating capacity of GSCs after hypofractionated radiation. Survival analyses in mice implanted by isogenic GSCs, either non-radiated (black curves) or after fr-IR (red curves).

**Figure 5 cancers-12-00570-f005:**
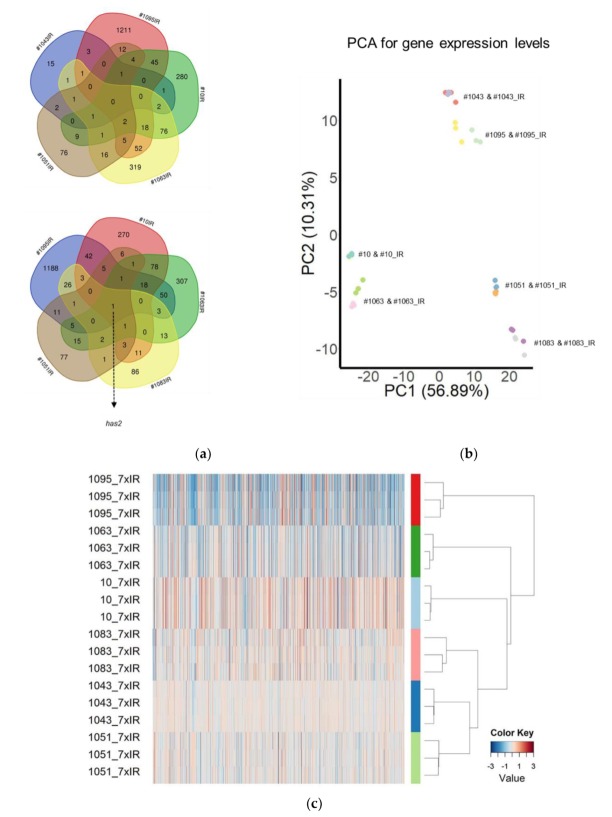
Diverse patterns of transcriptomic changes induced by hypofractionated radiation. (**a**) Venn diagram showing the overlap in differentially expressed genes (DEGs) identified in different GSC/GSC_IR pairs; (**b**) Principal component analysis of gene expression compared between isogenic GSC and GSC_IRs. Each line was analyzed in triplicate. Values aligned with axes show proportion of variance in percent for principal components 1 (PC1) and 2 (PC2), respectively; (**c**) Heatmap for log_2_ fold change (FC) of gene expression levels calculated for 1989 significantly altered genes (FDR < 0.05, log_2_(FC) ≥ 1).

**Figure 6 cancers-12-00570-f006:**
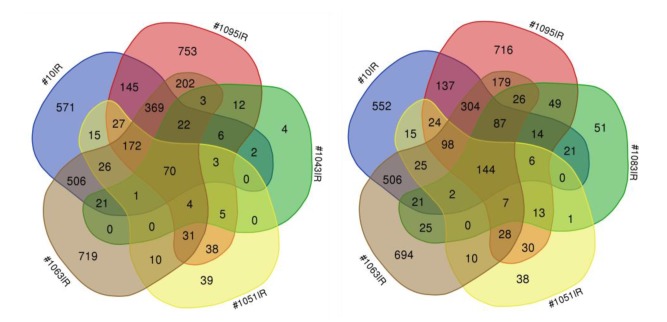
Gene ontology analysis of transcriptomic changes induced by hypofractionated radiation. Venn diagram showing the overlap in GO terms of differentially expressed genes (DEGs) identified through comparison of isogenic GSC and GSC_IRs.

**Figure 7 cancers-12-00570-f007:**
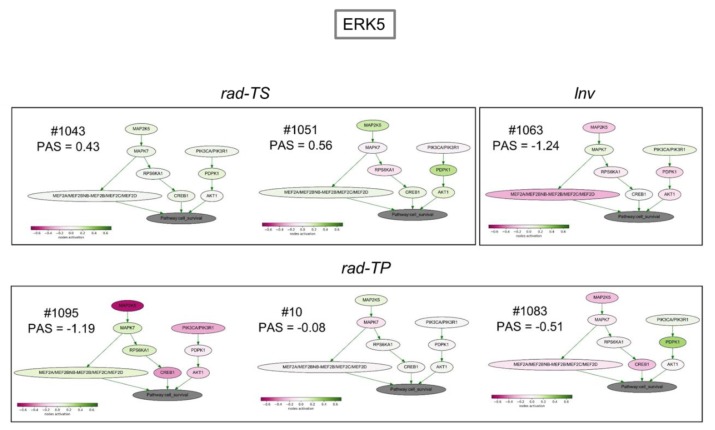
Activation profiles for signaling pathways Erk5, mitochondrial apoptosis and TGF-beta that were changed significantly in all GSCs/GSC_IR pairs tested. Numbers correspond to pathway activation scores (PAS) determined for individual pairs.

**Figure 8 cancers-12-00570-f008:**
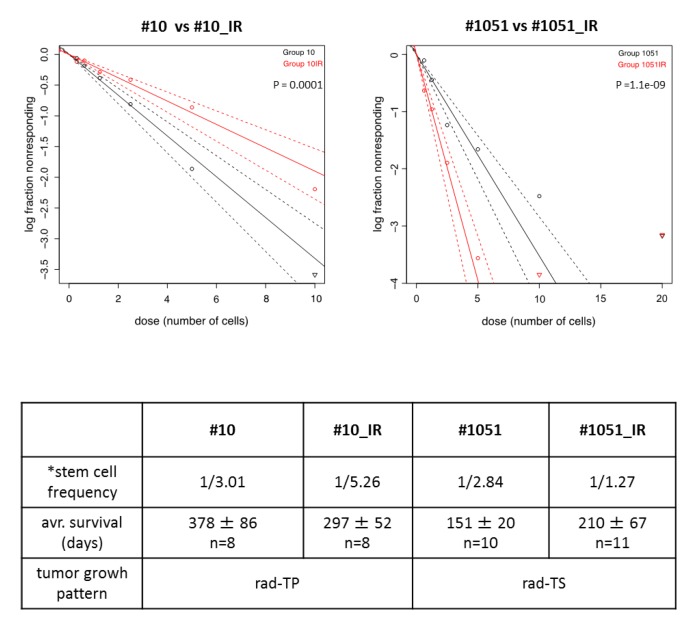
Comparative assessments of self-renewal in non-radiated and radiated GSCs by ELDA. Upper panels show the results for #10/#10_IR or #1051/#1051_IR pairs from three independent experiments. Table shows average frequency of stem cells (*) determined in three independent experiments. Also shown are average survival time of mice implanted with corresponding GSCs and tumor growth pattern after radiation.

**Table 1 cancers-12-00570-t001:** Lack of correlation between self-renewal and tumor-propagating capacity in human GSCs.

Line	GFAP Pattern	^1^ SCF	^2^ TGR	Ki67*^in vivo^* (%)
#10	GFAP*^Ind^*	1/3.0	378 ± 86 (*n* = 8)	8.73 ± 1.56
#1095	GFAP^Ind^	1/3.8	320 ± 96 (*n* = 7)	16.1 ± 6.98
#1063	GFAP^Ind^	1/3.5	226 ± 29 (*n* = 6)	12.7 ± 5.52
#1051	GFAP^Ind^	1/2.8	151 ± 20 (*n* = 11)	17.6 ± 6.22
#1080	GFAP^Const^	1/2.9	90 ± 21 (*n* = 4)	36.51 ± 11.77
G112SP	GFAP^Const^	1/2.9	89 ± 37 (*n* = 6)	23.21 ± 11.64
#1043	GFAP^Const^	1/30.0	116 ± 27 (*n* = 5)	18. 11 ± 7.45
#1083	GFAP^Const^	1/92.0	201 ± 32 (*n* = 7)	12.93 ± 3.77

^1^ “SCF”, stem cell frequency determined by ELDA [23] ^2^ “TGR”, tumor growth rates; Ki67 indices were determined by immunofluorescence staining of xenograft tumors.

**Table 2 cancers-12-00570-t002:** Proliferative changes invoked by radiation in different types of GSCs.

Isogenic Pairs	BIR (%) Self-Renewal	BIR (%) Differentiation	GFAP/TPC
#1063 #1063_IR	62.26 ± 6.89 (*p* = 0.007) * 86.4 ± 2.12	56.0 ± 1.59 (*p* = 0.02) * 6.0 ± 1.25	GFAP*^Ind^*/inv
#1051 #1051_IR	57.83 ± 15.63 (*p* = 0.26) 69.65 ± 8.65	14.09 ± 3.82 (*p* = 9.09 × 10^−8^) * 1.16 ± 1.32	GFAP*^Ind^*/rad-TS
#1095 #1095_IR	80.0 ± 1.89 (*p* = 0.78) 78.7 ± 2.97	1.4 ± 0.29 (*p* = 0.01) * 15.6 ± 3.23	GFAP*^Ind^*/rad-TP
#10 #10_IR	55.1 ± 3.28 (*p* = 0.12) 45.7 ± 2.11	21.6 ± 1.65 (*p* = 0.05) * 50.6 ± 2.59	GFAP*^Ind^*/rad-TP
#1043 #1043_IR	83.6 ± 8.05 (*p* = 0.003) * 51.2 ± 4.76	61.13 ± 12.52 (*p* = 0.2) 58.66 ± 14.30	GFAP*^Const^*/rad-TS
#1083 #1083_IR	53.92 ± 5.54 (*p* = 0.008) * 80.47 ± 11.75	32.96 ± 4.61 (*p* = 0.48) 37.39 ± 5.95	GFAP*^Const^*/rad-TP

“BIR”, BrdU incorporation rate; “GFAP”, GFAP expression pattern; “TPC”, tumorigenicity change pattern; *, statistically significant differences.

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
