# Peer review of "Diversity of Clinically Relevant Outcomes Resulting from Hypofractionated Radiation in Human Glioma Stem Cells Mirrors Distinct Patterns of Transcriptomic Changes"

_cancers, 2020, doi:10.3390/cancers12030570_

Round 1
Reviewer 1 Report
On one hand and other hand words style in sentences 73, 74, 287 and 288 can be improved.
Animal model radiotherapy transcriptomic expressions may not reflect the real world patient glioma stem cells patterns. However, for the technical correctness novelty I accept the paper for publication.
Author Response
We thank the Reviewer for his/her positive comments and suggestion to improve words style
Sentences specified by the Reviewer have been modified
Reviewer 2 Report
It is suggested to briefly discussed the effect of Erk5, TGF-beta and mitochondrial apoptosis pathways on CSC properties in the discussion. Besides, it may be better to conduct at least a short study showing that combined treatments targeting proliferation, apoptosis or neural differentiation with radiation could interfere with the radioresistance.
Author Response
We thank the Reviewer for the thoughtful and constructive comments, which are addressed below:
“It is suggested to briefly discussed the effect of Erk5, TGF-beta and mitochondrial apoptosis pathways on CSC properties in the discussion.”
Response: We thank the Reviewer for this suggestion. The effects of Erk5, TGF-beta and mitochondrial apoptosis pathway have been discussed in detail (pp. 14/15 (312-329)
“Besides, it may be better to conduct at least a short study showing that combined treatments targeting proliferation, apoptosis or neural differentiation with radiation could interfere with the radioresistance”
Response: We totally agree with the Reviewer that experimental testing the effects of combined treatments targeting proliferation, apoptosis and neural differentiation should be the next logical step following from this our study’s findings. However, this per se is a big project demanding extensive investigations at the molecular, cellular and tissue levels in vitro and in vivo. In fact, such investigations are currently ongoing in our lab. However at this point, the results are still too preliminary and need in-depth validation that cannot be accomplished within the timeframe of this manuscript revision.
Reviewer 3 Report
In this study, Kalasauskas et al. attempt to study the common effects of irradiation in GSC lines using in vitro and in vivo models. The study lacks rigor in both experimental design and statistical approaches. I summarized my major and minor comments:
Major:
-Figures 1 and 2 should be combined into 1 panel. Flow cytometric methods to quantify Ki67 and GFAP positivity are a much more appropriate method to quantify percent positivity in culture models. This will also easily exclude any dead cells to give a more accurate number.
-Table 1: In this table, the authors discuss a lack of correlation. No correlative methods are applied here. The authors should perform pearson correlation and provide the r value as well as a p value.
-Figure 3: Your GFAP positive cells appear to resemble radial glial cells. More markers should be used to show convincingly that these are astrocytes. GFAP, as well as Nestin, CD133, and other markers of radial glia should be used.
-Figure 4: Irradiation should be performed in vivo. I don't understand the importance of using mice in this experiment. If the point is to show that the cells grow slower after irradiation, a simple growth curve should have been done. Irradiation in vivo will get the authors to a closer measure of the effects of irradiation on a tumor in the brain, as this affects healthy neighboring cells and may trigger an immune response as well.
-Figure 5: (a), a heat map should be shown at a reasonable cutoff value to show the extent of transcriptomic changes. It is unclear what cutoff value the authors used, and what FDR ratio.
(b) Unclear what the percent values (10.13% etc..) next to PCA mean. is PCA here based on only differential genes?
-Figure 6: A venn diagram discribing the different gene ontologies is a more appropriate way to show the signaling pathways altered.
-Figure 7:no statistical method has ben applied to show any difference.
-Minor: English editing required all throughout.
Author Response
Reviewer 3:
”Figures 1 and 2 should be combined into 1 panel”.
Response: We thank the Reviewer for this logical suggestion. Figures 1 and 2 have been merged into one figure and are now shown in panels ”a” and ”b” of Figure 1.
“Flow cytometric methods to quantify Ki67 and GFAP positivity are a much more appropriate method to quantify percent positivity in culture models. This will also easily exclude any dead cells to give a more accurate number.”
Response: Cell lines used in the current study have already been analyzed extensively by flow cytometry in our previous studies concerned with the phenotypic evaluation of evaluation of a range of markers including GFAP. This is mentioned in the Results section (p.3, 101). Although we agree with the Reviewer on the appropriacy of flow cytometry we would like to point out that immunostaining is not only a widely accepted method to quantify the percentage of positivity in experimental samples, it also offers a number of advantages including the suitability of immunostaining for both cultured cells and formalin-fixed tissues. One of the prime objectives of our study was to compare same markers between cultured cells and tumor xenografts. As formalin-fixed
tissues are unsuitable for flow cytometry application, we chose immunostaining as the preferred method in order to minimize variability due to different techniques.
“Table 1: In this table, the authors discuss a lack of correlation. No correlative methods are applied here. The authors should perform pearson correlation and provide the r value as well as a p value”
Response: We thank the Reviewer for this suggestion, which has helped strengthening our conclusion about the apparent lack of correlation. Pearson correlation test has been performed. The results, r and p values are shown in Figure 2.
“Figure 3: Your GFAP positive cells appear to resemble radial glial cells. More markers should be used to show convincingly that these are astrocytes. GFAP, as well as Nestin, CD133, and other markers of radial glia should be used.”
Response: We would like to stress that we have never claimed on our manuscript that GFAP positive glioma stem cells characterized in our study are astrocytes or radial glia. Such a claim would also be incorrect because astrocytes and radial glia by definition are non-neoplastic cells. Although glioblastoma stem cells investigated in our study do share some morphophenotypic similarities with astrocytes they are either astrocytes or radial glial cells. For this reason we have used the term “astrocyte-like” (p.3, 113). To further emphasize the fundamental difference between glioma stem cells and normal astrocytes/glia we added a notion that glioblastoma stem cells are “reminiscent” of astrocytic progenitors or mature astrocytes (p.3, 104; p. 4,126).
“GFAP, as well as Nestin , CD133, and other markers of radial glia should be used”
Response: All GSCs as well as GSC xenografts have been assessed for GFAP and Nestin. Data are shown in Table S1, Fig. 3 and Fig. S2. While revising the manuscript we have realized that the legend to Figure S2 was missing in the original version. We apologize for overlooking this during previous submission. The description of nestin patterns in xenografts is now provided in the text (p. 7, 178/179) and legend to Fig. S2.
“…CD133, and other markers of radial glia should be used”
Response: All the lines used in this study have already been extensively characterized for CD133 and neural lineage markers in our previous studies (Barrantes-Freer 2012, Barrantes-Freer et al 2015) 2015). To avoid redundancy, we prefer not to show the data that have already been published and provide reference to the corresponding papers (p. 3, 101, refs. 19,20).
“Figure 4: Irradiation should be performed in vivo.”
Response: One of the prime objectives of our study was to correlate radiation-induced changes in gene expression with radiation-induced changes in the tumor-propagating potential. Addressing this issue requires using same batches of cells in gene expression analyses and tumorigenicity tests. This in turn necessitates that cells radiation should be performed prior to implantation to exclude the impact of transcriptomic differences related to spontaneous changes in the cellular state during tumor growth (Figure 3).
” I don't understand the importance of using mice in this experiment. If the point is to show that the cells grow slower after irradiation, a simple growth curve should have been done”
Response: One of the prime objectives of our study was to compare the tumor-promoting potential between non-radiated and radiation-treated glioblastoma stem cells. To that end, measuring the rate of tumor growth is unarguably the most straightforward approach because it enables a direct determination of the tumor-propagating capacity. In contrast, growth curves determined in vitro can be used as a surrogate measure to predict the tumor-propagating potential but only under the premise that cells proliferate at a more or less constant rate in vitro and in vivo. This is however not the case with glioblastoma stem cells, which can undergo considerable and highly variable changes in proliferation rates depending on a particular cellular state as shown in Figure 1b and Table 2.
Further, determination of growth kinetics of stem cells propagating in vitro requires them to be in the state of self-renewal. In the light of our findings indicating that some types of glioblastoma stem cells can undergo spontaneous differentiation in vivo (Figure 3a) determination of growth curves in vitro may not adequately reflect growth kinetics in vivo and lead to misleading conclusions.
“Irradiation in vivo will get the authors to a closer measure of the effects of irradiation on a tumor in the brain, as this affects healthy neighboring cells and may trigger an immune response as well”
Response: We do not share the view that in vivo irradiation would be more advantageous because it enables to recapitulate immune responses in the brain. Athymic mice used in our study are heavily immunocompromized and have the totally different tumor immune microenvironment compared with humans. In fact, the inability to simulate the in vivo immunity is considered a major disadvantage of immunodeficient mice.
“Figure 5: (a), a heat map should be shown at a reasonable cutoff value to show the extent of transcriptomic changes. It is unclear what cutoff value the authors used, and what FDR ratio”
Response: A heatmap with cut-off values FDR < 0.05, log2 ≥ 1 has been prepared and shown in Figure 5c.
“(b) Unclear what the percent values (10.13% etc..) next to PCA mean. is PCA here based on only differential genes?”
Response: We thank the Reviewer for pointing out that description of the data shown in Fig. 5 was insufficiently clear. Missing informations have now been provided in the legend to Fig. 5.
“Figure 6: A venn diagram discribing the different gene ontologies is a more appropriate way to show the signaling pathways altered.”
Response: Venn diagram for Gene Ontology terms has been prepared and shown in a new Figure 6.
“Figure 7: no statistical method has been applied to show any difference.”
Response: ELDA used to compare self-renewal is a statistical method for estimating depleted and enriched populations in stem cell assays. Mathematical basis of the ELDA method is described in Hu and Smyth 2009 (ref. #24).
“Minor: English editing required all throughout” Response: English editing has been done.
Reviewer 4 Report
The work is interesting and some aspects are innovative. The research project and the experimental design are appropriate, the conceptional structure is well organized. In the present form, the manuscript can be accepted with minor changes.
Point by point
FIGURES
In several figures (immunofluorescences) you should show the scale bar (e.g. 1a, 3a, 3b and supplementary).
MATERIALS AND METHODS
In the “Cell culture and cell based assays” subsection, you should indicate if, when and how cell line was last authenticated, maximum number of passages before the cells were analyzed.
Author Response
Dear Reviewer,
thank you very much for the encouraging words and appreciation of our work
we appreciate your suggestions, which we find most helpful to improve our manuscript
we have modified our manuscript according to your suggestions as follows:
FIGURES: scale bars have been indicated in Figs. 1a, 3a, 3b and Supplementary Figures
MATERIALS AND METHODS: Thank you for pointing out the lack of Information on cell lines authentication. This Information is essential indeed and we provide it now in the "Cell culture and cell based assays" sub-section of Materials and Methods (p. 16, 345/347)
Round 2
Reviewer 3 Report
Response:" Cell lines used in the current study have already been analyzed extensively by flow cytometry in our previous studies concerned with the phenotypic evaluation of evaluation of a range of markers including GFAP. This is mentioned in the Results section (p.3, 101). Although we agree with the Reviewer on the appropriacy of flow cytometry we would like to point out that immunostaining is not only a widely accepted method to quantify the percentage of positivity in experimental samples, it also offers a number of advantages including the suitability of immunostaining for both cultured cells and formalin-fixed tissues. One of the prime objectives of our study was to compare same markers between cultured cells and tumor xenografts. As formalin-fixed tissues are unsuitable for flow cytometry application, we chose immunostaining as the preferred method in order to minimize variability due to different techniques."
-Immunostaining is acceptable as long as it is done properly. There is no notion of how many cells are analyzed here, in how many fields, under blinded or double blinded conditions. Flow cytometric analysis is required for the conclusions presented. Figure 1 describes GSCs so I am not sure why the authors discuss formalin fixed tissues in their response. Clarity and rigor is lacking in this manuscript.
Response: We thank the Reviewer for this suggestion, which has helped strengthening our conclusion about the apparent lack of correlation. Pearson correlation test has been performed. The results, r and p values are shown in Figure 2.
-Figure 2 does not present the statistical analysis required. The authors should show an xy dot plot of all their datapoints with the trend line.
Author Response
Reviewer: “Immunostaining is acceptable as long as it is done properly. There is no notion of how many cells are analyzed here, in how many fields, under blinded or double blinded conditions.”Response: We thank the Reviewer for pointing out that the description of immunostaining-based evaluations has been insufficiently detailed. This critique has been addressed by providing a more detailed description of immunocytostaining experiments and illustratory examples shown in a new panel (a) of Figure 1. Information about number of cells, number of fields analyzed and controls for antibodies specificity has been provided in the Methods section (p. 16, 351-356) and legend to Figure 1a. Regarding the Reviewer’s request to provide information about “blinded or double-blinded conditions” we are not sure what the Reviewer means. Having analyzed the literature we could not identify studies that would define “double or double-blinded condition” as a criteria for immunostaining of cultured cells.
Reviewer: “Figure 2 does not present the statistical analysis required. The authors should show an xy dot plot of all their datapoints with the trend line.”Response: Figure 2 has been modified accordingly.
Reviewer: “Figure 1 describes GSCs so I am not sure why the authors discuss formalin fixed tissues in their response.”Response: In the current study, one of the prime objectives was to determine if the inherent capacity to differentiate can be realized in vivo during tumor growth driven by glioma stem cells. To be able to address this question, the inherent ability to differentiate has to be first determined under well-defined conditions for differentiation. Therefore, GFAP patterns established in vitro are not only essential, they provide the basis for interpretations of GFAP expression patterns in tumor tissues. As immunocytochemistry is suitable to analysis of both cells and tissues this method offers enables to directly compare GFAP patterns in vitro and in vivo. Another important advantage of immunocytochemistry is that it enables to analyze not only the proportion of positive cells but also the morphology of cells. This consideration is of particular importance for interpretations of GFAP patterns in terms of differentiation because variations in GFAP expression per se may not be related to differentiation if they are unaccompanied by changes in cell morphology, a hallmark of cell differentiation. In order to determine if the quantitative changes in GFAP expression are accompanied with the acquisition of differentiated morphology immunocytochemistry is the method of choice. To name but a few studies, immunocytochemistry has been utilized to evaluate the differentiation potential in glioma stem cells in a pioneer study by Singh et al 2003 CancerRes and more recent studies (Liu et al 2006 PNAS, Silvestre et al 2011 StemCells , Sharifi et al 2013 Cell Tissue Res, Bhat et al 2013 CancerCell, Schmidt-Edelkraut et al 2013 StemCells, Chao et al 2015Brain Behav).
Reviewer: “Flow cytometric analysis is required for the conclusions presented.”
Response: Compared with the visual analysis capabilities of microscopy flow cytometry is less informative with regards to the morphological aspects, which are essential for the context of the current study. Besides, all the lines used in the current study have been analyzed extensively by flow cytometry in our previous studies concerned with the phenotypic characterization for a range of lineage markers including GFAP. These data have been published as mentioned in the text and references (p. 3, refs. 19,20).
Some of these points have been partly addressed in our previous response and we hope, we have addressed this point more clearly now.